# Live tracking of moving samples in confocal microscopy for vertically grown roots

**Daniel von Wangenheim†, Robert Hauschild†, Matyáš Fendrych†, Vanessa Barone, Eva Benková, Jiří Friml\***

Institute of Science and Technology Austria, Klosterneuburg, Austria

**Abstract** Roots navigate through soil integrating environmental signals to orient their growth. The *Arabidopsis* root is a widely used model for developmental, physiological and cell biological studies. Live imaging greatly aids these efforts, but the horizontal sample position and continuous root tip displacement present significant difficulties. Here, we develop a confocal microscope setup for vertical sample mounting and integrated directional illumination. We present TipTracker – a custom software for automatic tracking of diverse moving objects usable on various microscope setups. Combined, this enables observation of root tips growing along the natural gravity vector over prolonged periods of time, as well as the ability to induce rapid gravity or light stimulation. We also track migrating cells in the developing zebrafish embryo, demonstrating the utility of this system in the acquisition of high-resolution data sets of dynamic samples. We provide detailed descriptions of the tools enabling the easy implementation on other microscopes.

**\*For correspondence:** jiri.friml@ist.ac.at

†These authors contributed equally to this work

**Competing interests:** The authors declare that no competing interests exist.

## Introduction

Root tips constantly explore the soil searching for water and nutrients. Their movement is propelled by cell division and elongation (*Beemster and Baskin, 1998*). The plant root tip comprises a stem cell niche giving rise to all cell types that build up the root. Root tips of *Arabidopsis thaliana* became an ideal model organ to study various aspects of developmental processes such as the control of the cell cycle, cell division orientation patterning, cell differentiation, cell elongation, cell polarity, gravitropism, hydrotropism, and hormone signaling (*Dolan et al., 1993*; *Malamy and Benfey, 1997*; *Friml et al., 2002*; *Moriwaki et al., 2014*). The temporal scale of these processes ranges from minutes (e.g. response to gravity) to hours (e.g. cell divisions) and days for cell differentiation (e.g. columella cell maturation). The rise of fluorescent live cell imaging and confocal laser scanning microscopy (*Stephens and Allan, 2003*; *Oparka, 1994*) enables the visualization of the dynamics of these processes with high spatio-temporal resolution. In practice, however, there is a trade-off between the resolution and the size of the field of view. This leads to the problem that a root growing in optimal condition will rush through the field of view in far less time than necessary to capture the process of interest. While the roots of 4–5 days-old plants grow up to 300 micrometers per hour (see below and *Beemster et al., 2002*), cytokinesis and cell plate formation require approximately 30 min (*Berson et al., 2014*; *Fendrych et al., 2010*), the complete cell cycle of transit-amplifying epidermal cells has been estimated to take between 10 and 35 h (Figure 6C and *Yin et al., 2014*; *Bizet et al., 2015*), whereas a maize cell located in the quiescent centre requires approximately 200 h to complete a cell cycle (*Clowes, 1961*). While the position of the root tip during short periods—timescales of minutes—can be corrected by registration of the images post-acquisition, longer events require repositioning of the microscope stage to keep the root tip in focus and within the field of view. Manual repositioning is an option, but it is not very convenient. Several automatic

solutions for root tip tracking have been published (*Campilho et al., 2006*; *Sena et al., 2011*); however, the authors did not provide a comprehensive documentation of the setups that were used.

Another very important aspect of plant live imaging is the control of environmental signals perceived by the plant, such as light, temperature, availability of nutrients, and gravity. The latter is constant and plant organs are influenced by the vector of gravity. During gravistimulation, the plant hormone auxin gets depleted from the upper side of the organ and accumulates in the lower side of the root, triggering the inhibition of growth. The root bends because the upper part of the root continues to grow (*Rakusová et al., 2015*). When gravitropic responses are the focus of the research, the position of the plant during live imaging becomes crucial. Most microscope setups keep the sample in a horizontal position, leading to a constant gravistimulation of plant organs. This feature was utilized to analyze calcium signaling after gravistimulation (*Monshausen et al., 2011*) by imaging the lower and upper sides of roots using either an upright or an inverted microscope setup in order to observe differences between these areas.

Nevertheless, horizontal sample positioning imposes serious limitations on any long-term imaging as the root continually attempts to bend downwards. In particular, as the orientation of the root with respect to the vector of gravity is fixed, it limits the way gravitropism can be studied.

Likewise, light, another crucial environmental cue for plant development, is usually not optimized for plant growth in conventional microscopes. Plant organs perceive light and show phototropic behavior. In fluorescence microscopy, light is used to excite fluorophores. This was harnessed by *Lindeboom et al. (2013)*, who used the excitation light as the stimulus to reorient the cortical microtubule array. Illumination of the leaves is crucial to keep the plant in a photosynthetically active state, in particular when performing long-term experiments. Optimal growing conditions are provided by a microscope setup that combines an illumination system with a vertical sample mounting, as has been established for light-sheet microscopy (*von Wangenheim et al., 2017*; *Maizel et al., 2011*; *Sena et al., 2011*). In confocal microscopy, vertical sample mounting was achieved with the introduction of a periscope tube in the optical path (*Monshausen et al., 2009*), however, this is at the expense of control of the motorized stage.

Here, we describe in detail a confocal microscope setup with vertical sample mounting and integrated illumination that we developed for the optimal imaging of *Arabidopsis* (*Figure 1*). We additionally developed a rotation stage that enables rapid gravistimulation while imaging with minimal

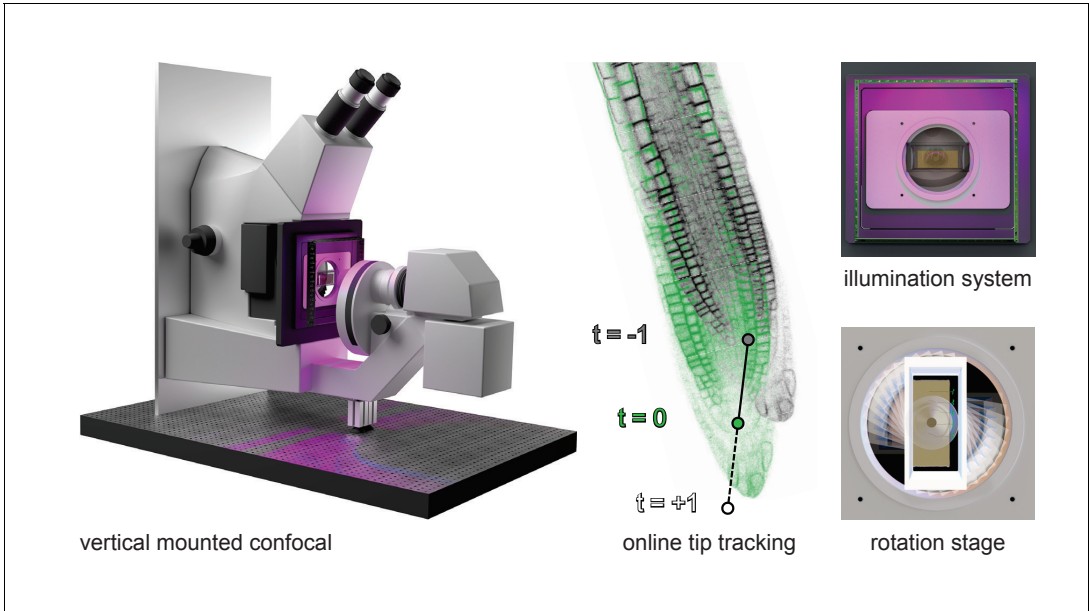

vertical mounted confocal     online tip tracking     illumination system     rotation stage

**Figure 1.** Overview of the vertical microscope setup. The vertically mounted confocal microscope enables long-term imaging of plant roots growing along the gravity vector under controlled illumination. Growing root tips are tracked using the TipTracker MATLAB program. By turning the sample around the optical axis using the rotation stage, gravistimulation is applied to the growing root tips.

disturbance of the plants. Further, we present TipTracker – a MATLAB-based program that enables the user to follow growing root tips or other moving samples and record time-lapse series over prolonged periods of time. TipTracker also outputs the coordinates of the individual root tips over time that can be used to reconstruct the trajectory of individual root tips and calculate their growth rates during the experiment. We demonstrate the performance of the system by several case studies focusing on root growth, cell division, cell lineage establishment and gravitropism. Additionally, we demonstrate that TipTracker can be used for any other type of specimen, for instance, Zebrafish embryos. TipTracker can interact with the graphical user interfaces of various commercial microscope software programs. We provide the source code for the implementation of TipTracker and

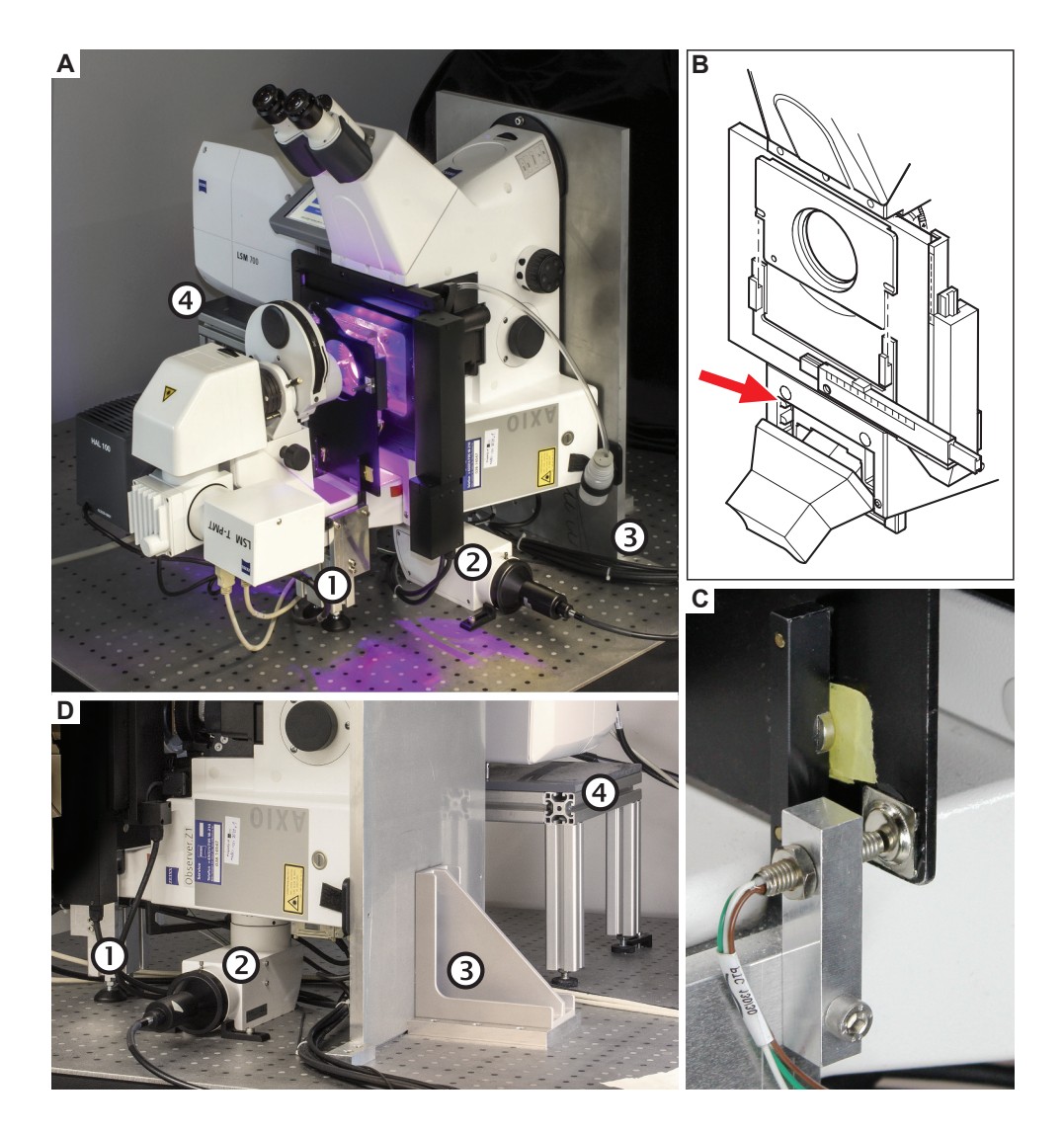

**Figure 2.** The vertically-mounted microscope setup. The microscope body (Zeiss Axio Observer) was rotated by 90° and mounted to an optical table using a strong angle bracket. The scan head (Zeiss LSM 700) has been raised but retained its original orientation. (**A**) Photograph of the setup in our laboratory. (**B**) The laser safety mechanism. Since the transmitted light arm can no longer be reclined/tilted, the laser safety shield would limit the access to the sample. The reed switch (red arrow) was removed, and the screws holding the safety shield were replaced by pins and magnets. The reed switch was relocated to the bracket that holds the shield, depicted in (**C**). (**D**) Photograph showing the stand (1), the 90° adapter for wide field fluorescence excitation (2), the aluminium plate and mounting bracket (3), and the scan-head table (4).

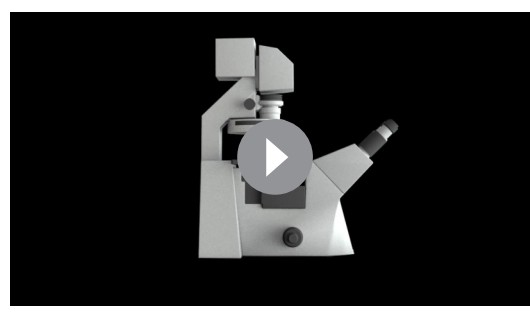

**Video 1.** 3D model of the vertical mounted confocal microscope.

associated scripts for two different microscope platforms (Zeiss LSM700, LavisionBiotech TriM Scope II), making it a versatile tool that can be easily modified and adapted.

## Results

### The vertical mounted confocal setup enables imaging of roots growing in a vertical position

An inverted confocal microscope (Zeiss Axio Observer with LSM 700 scanhead) was mounted on a 1 cm-strong aluminium plate (3D-CAD file provided in *Supplementary file 1*) and turned 90° so that the microscope is flipped onto its back side. A stand supports the transmitted light illumination carrier of the microscope (*Figure 2A, D*, and *Video 1*). Since the transmitted light illumination carrier could no longer be flipped back, two problems arose: (1) sample insertion became difficult and (2) the automatic laser shut-off was no longer working, thus laser safety was not guaranteed during sample exchange. To resolve this, we remounted the laser safety plate with magnets and connected the automatic laser shut-off mechanism to it (*Figure 2B,C*). Now, before inserting the sample, the user must remove the laser safety plate entirely in order to have free access, which turns the lasers off automatically. Furthermore, the standard fluorescent light coupling port could no longer be used and needed to be replaced with a 90° adapter. A detailed description of the required modifications is part of the supplementary documentation (*Supplementary file 1*). The scan head was raised, but retained its orientation. In this way, the functionality of the entire system is unaffected. A microscope that is suitable for turning 90° should have several features: The scan head needs to be connected to the side port of the microscope body. The flange that connects the scanhead to the body has to be rotationally symmetric. The distance between the scanhead and the microscopy body has to be large enough not to interfere with the stage or other components of the body. Microscopes that employ piezo focusing (stage or objective) could be problematic as piezos are sensitive to lateral forces. The microscope body needs to incorporate a z-drive that actively moves the objective down and does not rely on gravity for the downward motion. As far as we can tell, this is the case for all modern microscopes of all the major manufactures. If the z-drive features a position encoder and closed loop operation, then this is given.

Plants require light for photosynthesis and development. At the same time, directional illumination is a spatial clue that triggers plant tropisms. Therefore, to satisfy the seedlings' need for light and to enable directional photostimulation, we implemented a custom illumination system. The spectral quality and intensity matches the one in our growth room – LED illumination with blue maxima at 453 nm and red at 625 nm with intensity optimized for *Arabidopsis* (*Figure 3E,F*). Red and blue LEDs are arranged in a square facing the specimen that is mounted on the safety laser interlock plate (*Figure 3A–D*). Each side of the lamp can be switched on individually for directional illumination, and the intensity of illumination can be set to any value between 40 and 180 µmol m$^{-2}$s$^{-1}$ (*Figure 3F*). Initially we intended to shutter the light during image acquisition, but it turned out that when choosing proper filter settings, we do not detect any light coming from the LEDs (*Figure 3G, H*). This allows for keeping the illumination on during image acquisition.

In summary, we developed a vertical microscope setup with controlled illumination that enables multi-day live imaging. The system keeps the entire functionality of the CLSM microscope, including the motorized stage. This is essential for the acquisition of multiple positions during time-lapse experiments and also for the tracking of the root tips, as described below. One limitation of the system is that the usage of immersion objectives becomes complicated as the immersion solution flows down. We overcame this by using a viscous hair gel with a refractive index similar to water. The microscope is, however, mainly intended for the acquisition of multiple position and long time-lapse experiments, thus the use of any immersion liquid is problematic in a standard microscope setup as

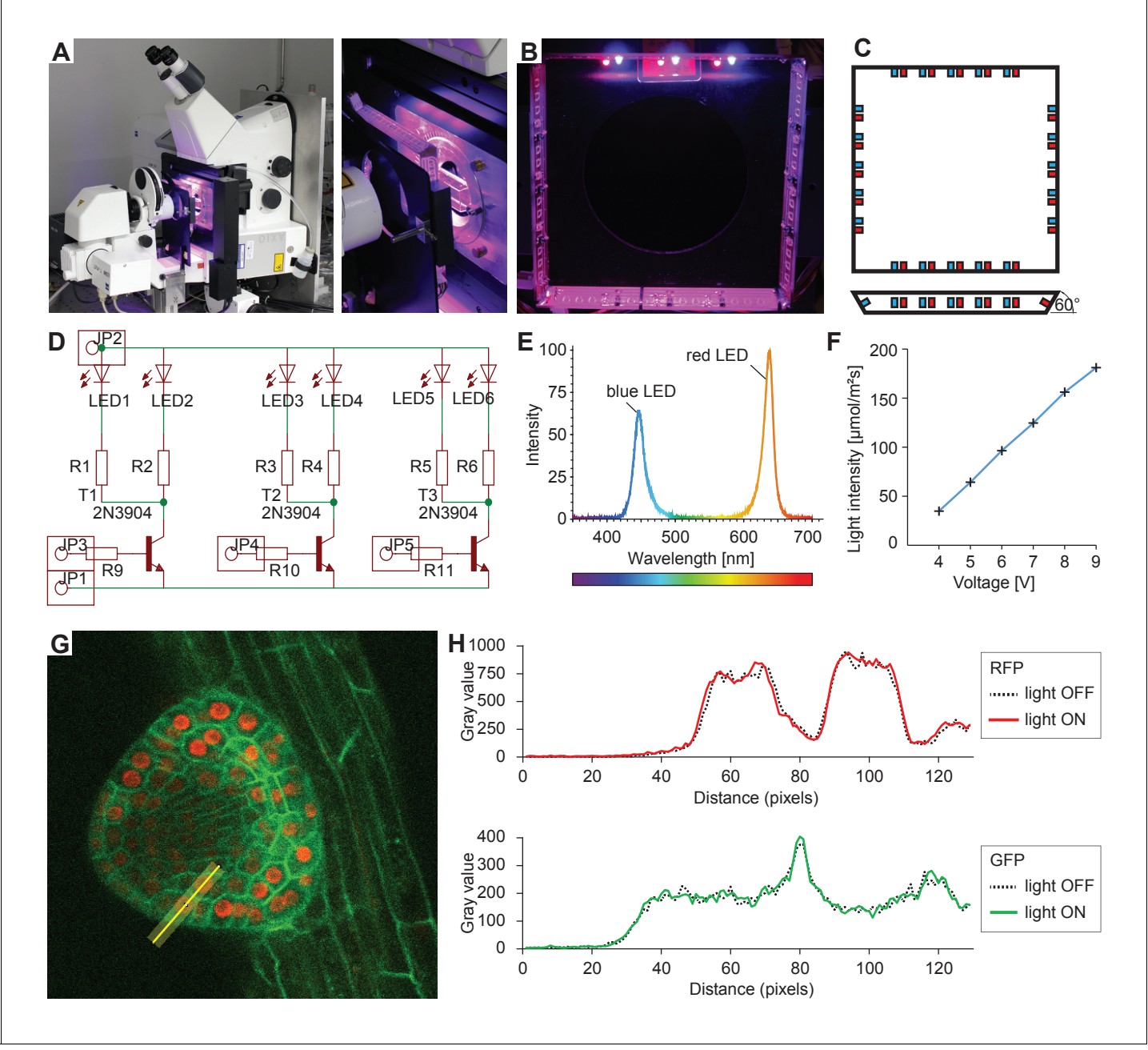

**Figure 3.** Integrated sample illumination setup. (**A**) Photograph of the LED illumination system attached to the microscope. Red and a blue LED are arranged in a square. Each side of the square can be switched on/off individually for directional lighting. (**B**) Photograph from the sample side. (**C**) Schematic of the LED square arrangement. Each side is tilted by 60° toward the sample. We provide the board design file in the ***Supplementary file 1***. (**D**) Schematic of the circuit diagram of one side of the lamp. LED: light-emitting diode, R: resistor, T: transistor, JP: pinhead. (**E**) The emission spectrum of the lamp. (**F**) The voltage can be adjusted in the range of 3.5–9.5 V. Resistors used to reach light intensities ranging from 40 to 180 µmol/m²/s: R1-8: 220 Ohm, R9-12: 1220 Ohm. (**G**) Single optical section recording of a lateral root primordium expressing GFP-plasma membrane marker UBQ10::YFP-PIP1;4 and RFP-nuclear UBQ10::H2B-RFP marker. Green fluorescence was collected between 490 and 576 nm, red fluorescence was collected between 560 and 700 nm. The fluorescence intensity profile of the yellow line (21 px width) is plotted in (**H**). (**H**) Intensity profile along the yellow line shown in (**G**) of the red and green channels with illumination system switched on or off, respectively, demonstrating that RFP/GFP imaging is not affected by the illumination.

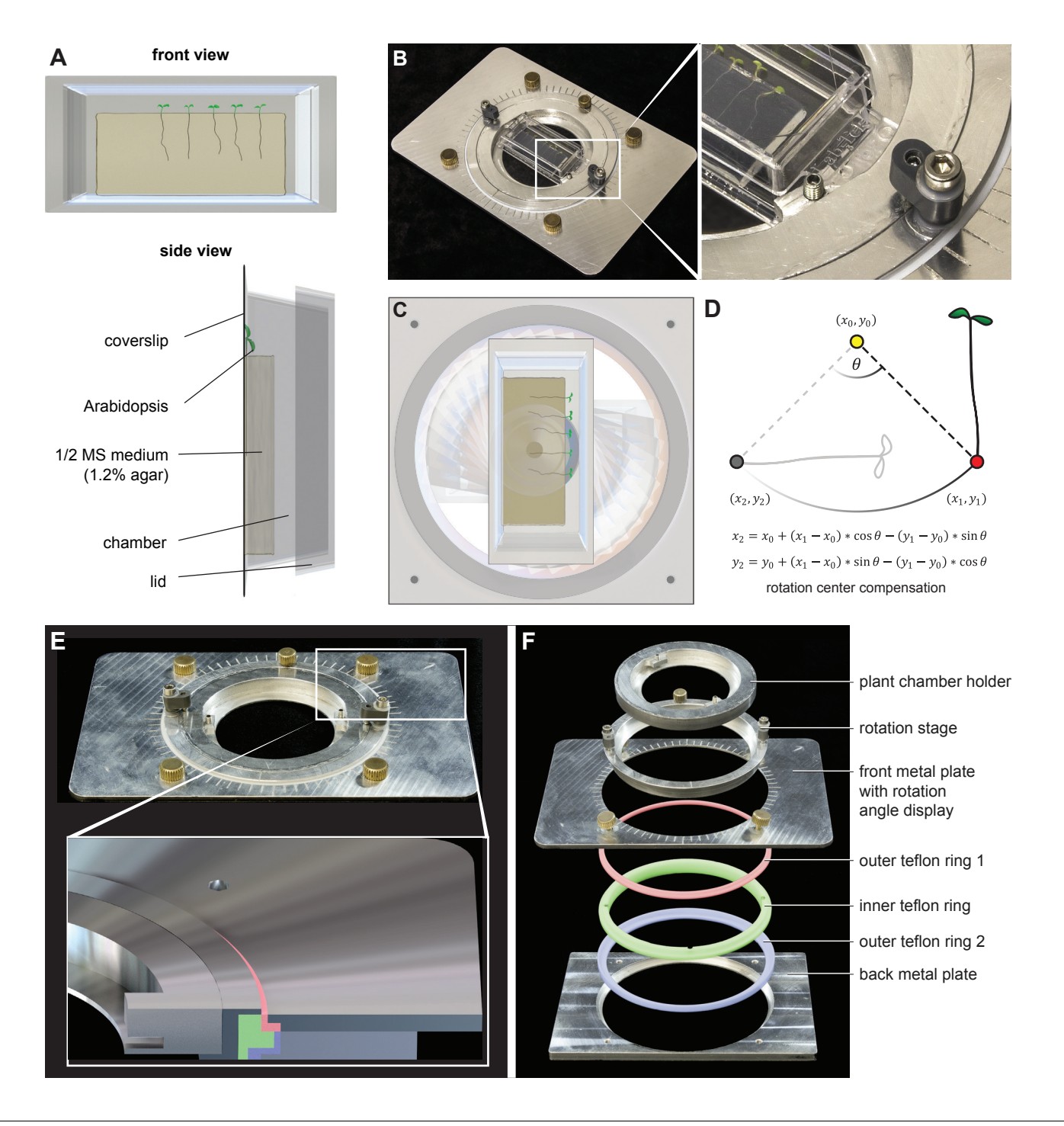

**Figure 4.** Gravistimulation of samples using the sample rotation stage. (A) Roots grow in chambered coverslips between the coverglass and a block of agar. (B) The sample chamber is placed in the rotation stage. The enlargement highlights the small screw mounting the sample chamber in the rotation stage, while the bigger screw fixes the rotation inset. (C) The chamber can be rotated around the optical axis inside the microscope inset which leads to a reorientation of the root with respect to the gravity vector. (D) The new positions of the root tips after rotations are calculated using a script to minimize the delay between rotations and imaging (***Supplementary file 2***). (E) and F) The construction of the sample rotation stage. The rotation stage is pressed in an inner Teflon ring (green ring), which is held together by an inner- and outer Teflon ring (red and blue) compressed by the front- and back aluminium plates. This arrangement provides smooth rotation of the sample. 3D files are provided in ***Supplementary file 1***.

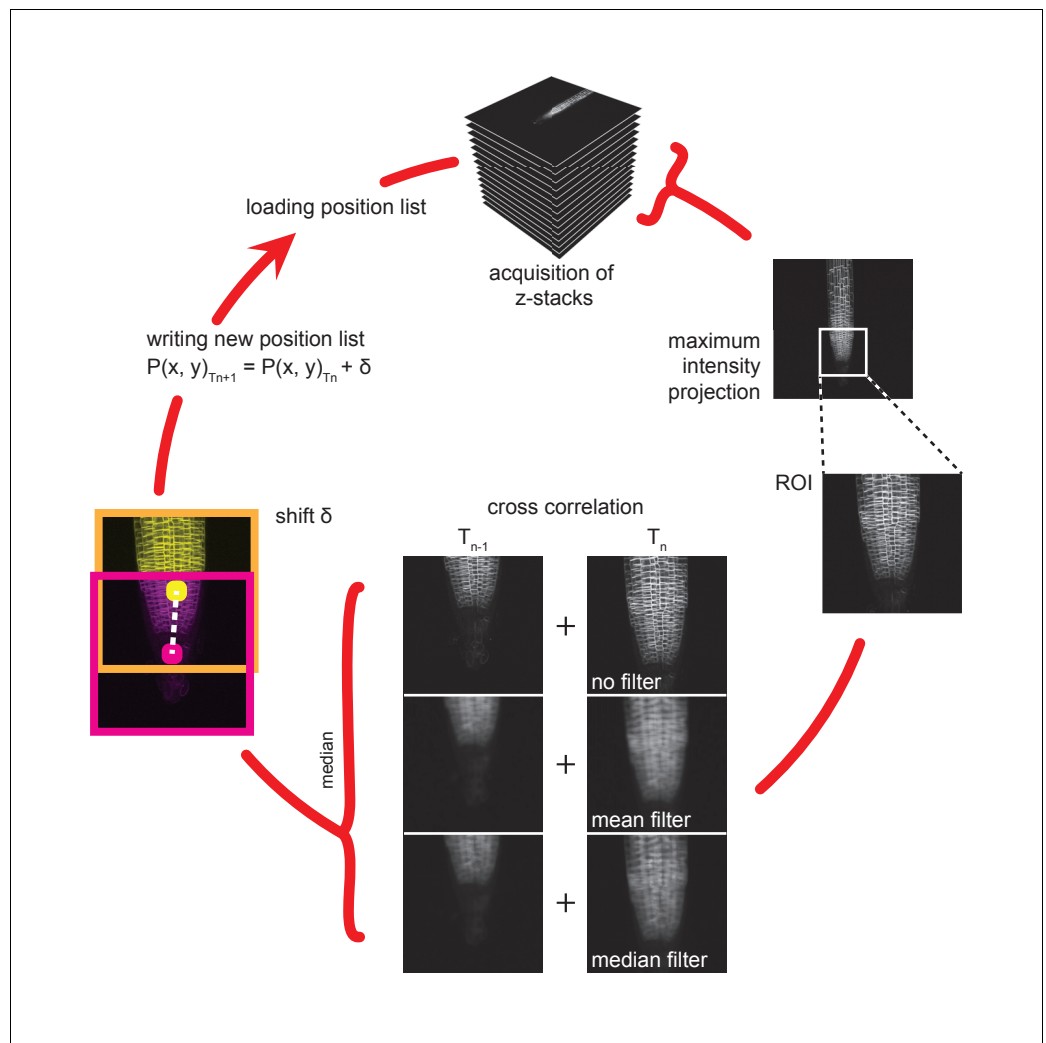

**Figure 5.** Working cycle of the TipTracker program. A region of interest is selected from a maximum intensity projection of a z-stack. Mean and median filters are applied. A direct cross-correlation is performed between the current and the prior time point on the differently filtered and non-filtered images. Note that this procedure is purely based on the similarity of the images within the region of interest and makes no assumption about the sample. The median of the three results is used as the shift to make the calculation more robust. The calculated shift is added to the current position and is used as a prediction of the position of the root in the subsequent time point. The new position list is saved and then loaded at the start of the next acquisition time step.

well. Therefore, for this type of experiments, we primarily employ a 20x/0.8NA or a 40x/0.95NA dry objective.

## Rapid gravistimulation experiments using the rotation stage

For gravistimulation experiments, it is important that the sample is securely fixed. Also for the long-term time-lapse experiments, the seedlings need to be well adapted to the experimental conditions. We grow the seedlings in a Nunc Lab-Tek chambered coverglass. While the roots grow between the glass and a block of agar, the cotyledons remain free to the air (*Figure 4A,B*). A detailed sample preparation protocol is provided in the Materials and methods section. Ideally, the sample preparation is performed a few hours before imaging to give the plants time to acclimate and recover from the stress of transfer. Plants can be cultured in the chambers for a period of several days. To test whether the roots grow well in such a setup, we compared the growth between the block of agar and the coverslip with the growth of roots in the same chamber on the surface of the agar block.

**Figure 6.** Time-lapse recording of eight *Arabidopsis* root tips expressing UBQ10::YFP- PIP1;4 over the course of 38 h. (**A**) Maximum intensity projections of a single time point for the eight roots tracked. (**B**) Growth rates of the root tips were calculated from the output of the TipTracker program. The yellow and grey areas indicate when the LED illumination was on or off, respectively. (**C**) Cell division and elongation are visualized for the root #5. Each new cell wall is highlighted so that the original cell walls are in yellow, the second generation of the walls is in cyan, the third generation is in magenta, and the fourth generation in green. The last image of the series is shown on the right side. A stack of 14 images (x/y/z: 1400 × 1400 × 14 pixels, voxelsize: 0.457 × 0.457 × 2.5 µm³) was captured every 20 min for a period of 38 h 20 min using the Plan-Apochromat 20x/0.8 air objective lens. Scale bars: (**A**) 100 µm, (**C**) 40 µm.

Roots between agar and glass grew 88% of the ones growing on the surface (87.94% + −10.5% STDEV, control 100% + −10.9% STDEV, n = 21 and 20, respectively), which can be attributed to the mechanical impedance of the setup.

Gravitropism of the root is a canonical example of the adaptation growth response to environmental stimuli. It involves the asymmetric distribution of the phytohormone auxin (*Went and Thimann, 1937*) and has been used to study the plant's perception of gravity (*Baldwin et al., 2013*), cellular polarity of the PIN auxin transporters (*Adamowski and Friml, 2015*; *Friml et al., 2002*), targeted protein degradation (*Baster et al., 2013*; *Abas et al., 2006*), and other signaling processes (*Shih et al., 2015*). The horizontal arrangement of objective lenses allows for mounting the sample in a vertical position; the roots can then grow down along the gravity vector. Since the gravity vector cannot be modified easily, we developed a microscope sample holder that can be rotated

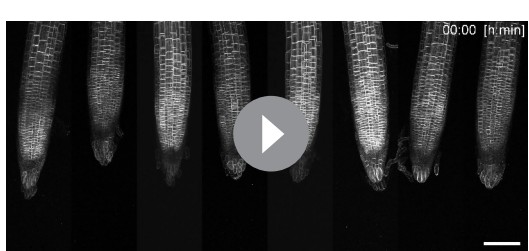

**Video 2.** Time series of eight *Arabidopsis* root tips recorded over 38 h.

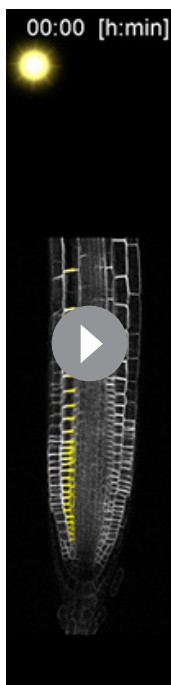

**Video 3.** Single slice of root tip number 5 of *Video 2*.

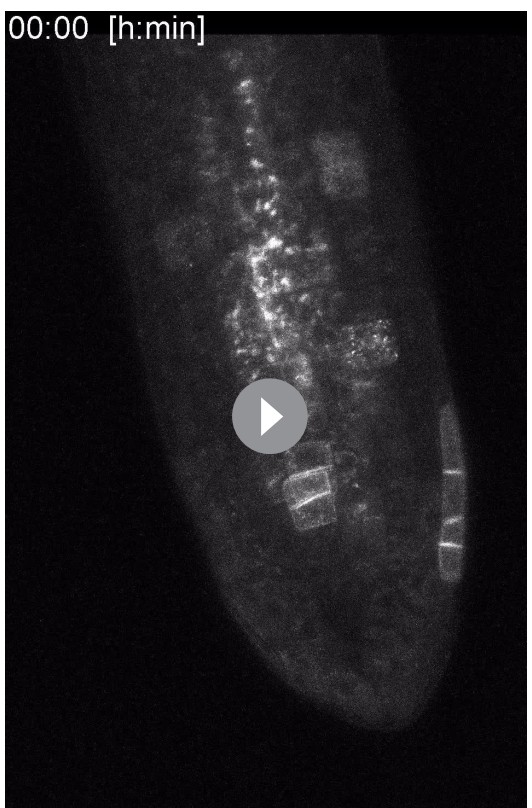

**Video 4.** High-resolution time series of a root tip expressing KNOLLE::GFP-KNOLLE.

by any degree around the axis of the light path (*Figure 4C*). In this way, the roots can be observed before and shortly after the gravistimulation, and due to the axis of rotation, the 'upper' and 'lower' root sides are equally accessible to imaging. The rotation stage is an aluminium frame with a rotating inset that holds the sample chamber. The inset and the frame are connected by a number of rings made of Teflon to provide smooth and precise sliding (*Figure 4E,F*). *Supplementary file 1* contains a 3D CAD file of the rotation stage designed for a motorized stage (Märzhäuser Scan IM). In order to minimize the time the user spends finding the roots after rotation of the inset, we developed a MATLAB-based script that calculates the new positions of the root tips (*Figure 4D*, *Supplementary file 2*). The experimental procedure was as follows: First, the motor coordinates of the mechanical centre of rotation had to be determined. To this end, the inset holding the sample chamber was replaced with a disk into which a small hole (diameter 200 μm) had been drilled, which coincides with the centre of rotation. The hole was centred in the field of view and the motor position was saved in a file. Then, the disk was replaced with the sample holder and the positions of the root tips were saved. After imaging the first part of gravistimulation experiment (roots in vertical position), the rotation was applied. The MATLAB script (*Supplementary file 2*) was executed, and output the new position of root tips. The mechanical precision was good enough that the calculated positions deviate only slightly from the actual ones and imaging could be continued within approximately 3 min after the rotation, which is the time needed to reposition and restart the time series.

Thus, our rotating stage enables the user to select any sequence of gravistimulations desired, and subsequently a very rapid image acquisition, providing the setup necessary for high-resolution studies of gravitropism.

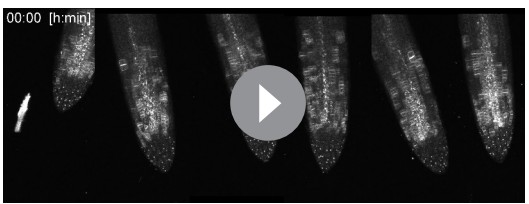

**Video 5.** 24 h recording of six *Arabidopsis* root tips expressing KNOLLE::GFP-KNOLLE.

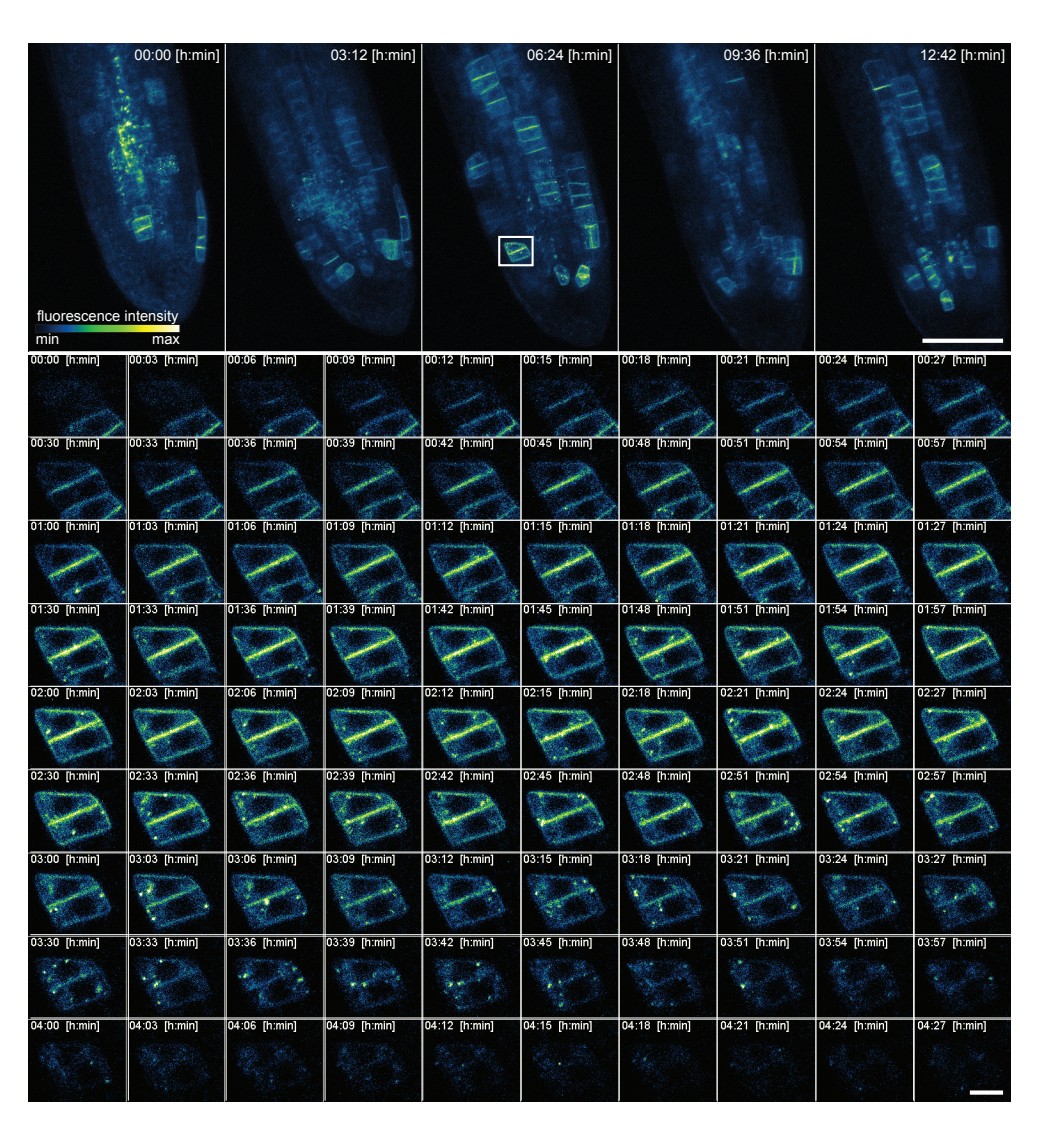

**Figure 7.** High spatio-temporal resolution time lapse recording of a 5-day-old *Arabidopsis* root tip expressing KNOLLE::GFP:KNOLLE. The upper panel shows the maximum intensity projection of five time points out of 255. The appearance and disappearance of the fluorescent fusion protein during cytokinesis of a single cell is depicted in the lower panel. A stack of 10 images (x/y/z: 1448 × 1448 × 10 pixels, voxelsize: 0.138 × 0.138 × 2 $\mu m^3$) was captured every 3 min for a period of 12 h and 42 min using the Plan-Apochromat 40x/0.95 air objective lens. Scale bars: 100 $\mu m$ upper panel, 10 $\mu m$ lower panel.

## TipTracker automatically recognizes and follows root tips during growth

A root tip of a 4–5 day-old *Arabidopsis* seedling grows approximately 50–300 µm per hour (see below). This means that it moves through the field of view of a 20x objective within 1–2 h. To be able to observe the root tips for a longer period of time, we developed the root tip-tracking program TipTracker. Importantly, TipTracker makes it possible to observe multiple samples independently over long periods of time regardless of the individual behaviour. Besides imaging, TipTracker outputs a file with the coordinates of the individual positions that can be used to measure the growth rate of the individual roots as well as to retrace the path of growth.

The acquisition of multiple channels, z-stack and positions is set up in the microscope's software, while the time series is managed by TipTracker, which also handles image file loading, motion

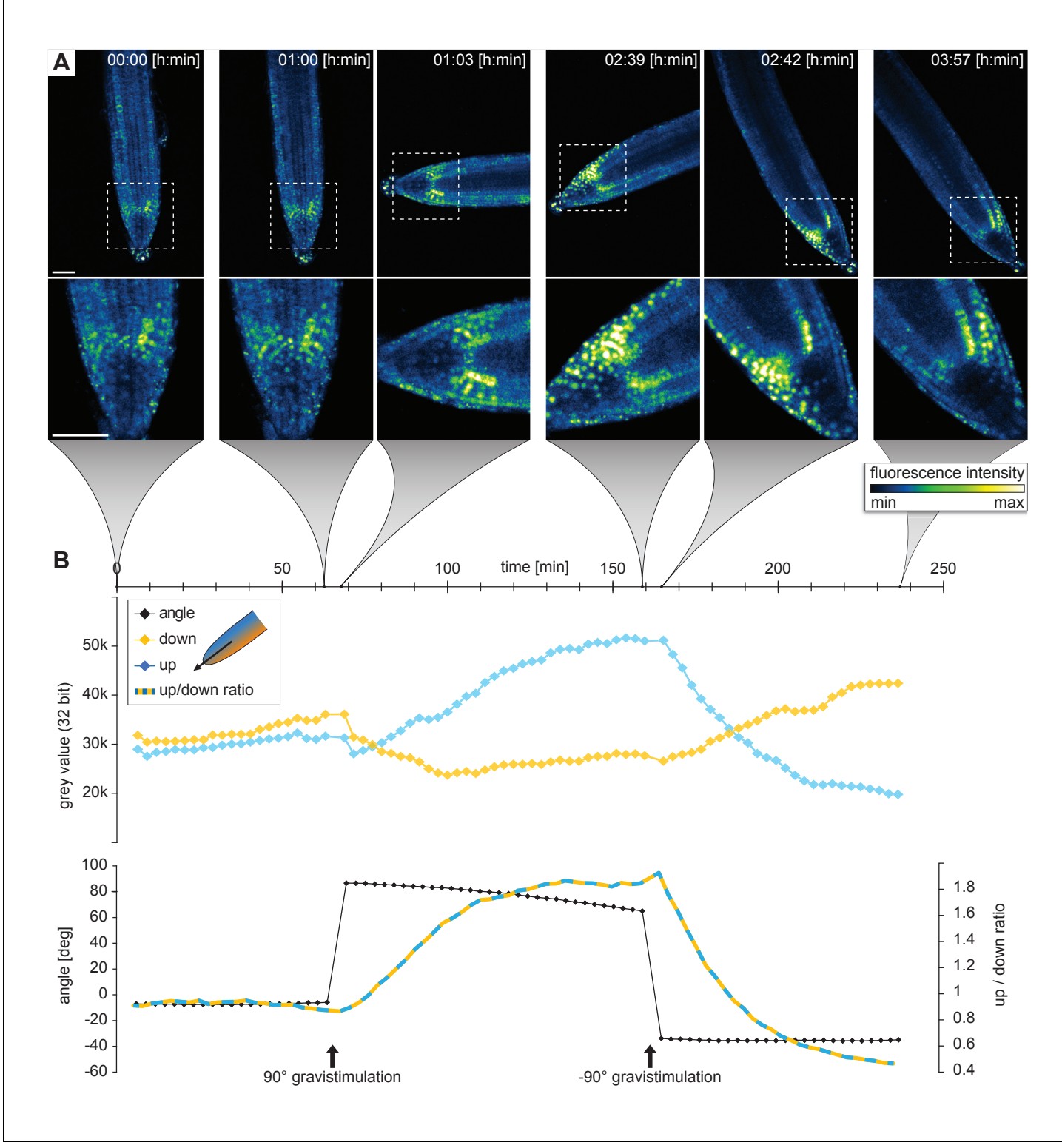

**Figure 8.** Recording of the gravitropic response of a 5-day-old *Arabidopsis* root tip expressing the DII-VENUS marker. Initially, the root growth was followed for 1 h in the vertical position. Subsequently, the plants were gravistimulated by a 90° rotation (clockwise) and imaged for 1 h 36 min. Finally, the plants were rotated back (90° counter clockwise) and imaged for another 1 h 15 min. Three roots were imaged simultaneously, see also the **Video 5**. (**A**) Sum intensity projections of six time points out of 80. (**B**) Upper diagram shows the fluorescent intensity on the upper/left side (blue line) compared the lower/right side of the root tip (yellow line). The lower graph shows the root tip angle and the ratio of fluorescence intensity upper /

*Figure 8 continued on next page*

*Figure 8 continued*

lower side. A stack of five images (x/y/z: 1024 × 1024 × 5 pixels, voxelsize: 0.625 × 0.625 × 3 µm³) was captured every 3 min for a period of 4 h using the Plan-Apochromat 20x/0.8 air objective lens. Scale bars: 50 µm.

tracking evaluation, the generation of the new position list, and displaying the root growth history. Once the images of a time point are saved a maximum intensity projection is generated for each specimen. A region of interest is cropped, filtered (mean and median) and compared to the corresponding image of the previous time point. The maximum of the direct cross-correlation of both time points yields a lateral shift $\Delta$ that is used to update the lateral sample displacement $\delta$. This method makes no assumptions about the shape or brightness of the samples or the type of movement and is thereby not limited to roots; it is, in fact, entirely independent of the specimen and can be used for all samples that move autonomously or through external forces. The growth of a root between two time points is then predicted by $\delta$ (t)= $\delta$ (t-1)+ $\Delta$. Finally, a new list $P(x,y)_{Tn+1}= P(x,y)_{Tn}+\delta$ of the predicted positions of the roots is generated and loaded into the microscope control software and the next acquisition is started (*Figure 5*). This process is repeated for each step in the time series. The growth kinetics of each root can then be derived from the history of the recorded positions. More elaborate position prediction methods such as autoregressive motion or Kalman filter-based approaches could easily be implemented, but proved unnecessary in the case of root growth, since the change in growth speed and direction is slow compared to the interval between two time steps. Likewise, tracking in 3D, which can be accomplished in a straightforward manner by cross correlating different slices from the stacks or by maximum intensity projections in the x- and y- directions, were also found to be redundant, as the roots are confined between the coverslip and the agar block.

TipTracker interacts with the microscope control software (e.g. ZEN) by generating mouse and keyboard inputs by means of AutoIt scripts, which is a freeware scripting language designed for automating the Windows GUI. This solution has crucial advantages: First, it can be used with any type of microscope system since it does not rely on vendor-supplied programming interfaces. Furthermore, both experts and non-experts can easily adapt the system from one setup to another (e.g. ZEN Black -> ZEN Blue). We used TipTracker mainly in conjunction with the Zeiss ZEN2010 software controlling the LSM700 inverted microscope, but we have also successfully implemented TipTracker on an upright two-photon microscope (LaVision Biotech TriM II) to track the movement of cells within the enveloping layer during zebrafish epiboly. This also demonstrates that the tracking algorithm is not limited to roots and can be used for all moving samples. In order to exemplify the modifications necessary to adapt TipTracker to a new platform, we also included this version in *Supplementary file 2*. For each new system, the AutoIt scripts must be modified to correctly interact with the crucial control buttons of the software GUI, and we provide an illustration of this process within the commented AutoIt scripts (*Supplementary file 2*). We also provide a compiled version of TipTracker for usage with Zeiss ZEN2010 (*Supplementary file 2*). When the ZEN-based setup is used, separate images for each time point are saved in the lsm file format, each containing multi-position, multi-color z-stacks. We provide a script for the open-source software Fiji (*Schindelin et al., 2012*) that converts these multi-position files into multiple hyperstacks, each containing a single position (*Supplementary file 2*).

The program is designed to follow actively growing root tips in a highly efficient manner, as we demonstrate below. In case the tracking algorithm loses a sample, this can result in excessive stage movements. In order to protect the objectives, we implemented a limit on the maximum degree of stage movement. When one of the positions exceeds this user-defined

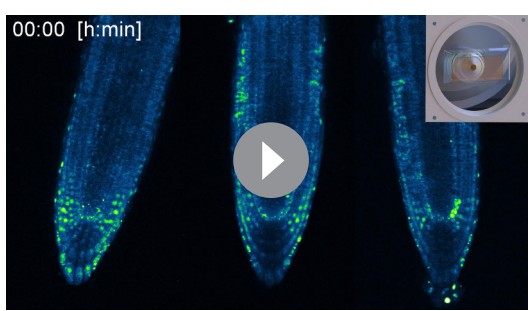

00:00 [h:min]

**Video 6.** Gravistimulation experiment. Time series of three *Arabidopsis* root tips expressing the auxin response marker DII-Venus.

limit, the tracking of that particular position is stopped, while the other positions are further tracked.

Limitations are that the computer should not be interfered with during imaging, as this could confuse the communication between TipTracker and the imaging software. In addition, online tracking with TipTracker creates a time overhead compared to a time series that is acquired directly with the microscope control software, since at each time point the acquisition is stopped, the data stored and then read again.

In summary, our root tip-tracking program TipTracker allows for online long-term tracking of root tips or other moving samples and can be easily implemented on a wide range of microscopes.

## Biological examples

### Long-term imaging and tracking of root tips

To test the ability of the TipTracker software, we imaged roots expressing the plasma membrane marker UBQ10::YFP-PIP1;4 over a period of 38 h with an imaging interval of 20 min. We imaged a 14-slice z-stack of eight roots for 116 imaging cycles (*Figure 6A*, *Video 2*). The program successfully tracked all roots. We coupled the illumination system to a regular time switch to simulate day and night. The growth rate of individual roots varied from 50 to 250 micrometers per hour, but the rate of all roots dropped during the night period and increased again in the day period (*Figure 6B*). In the resulting images, the cell division in the meristematic zone and the progression toward the transition and elongation zones can be observed (*Figure 6C*). We took a single plane of one of the datasets, cut out a small area overlapping with one of the cell files (cortex) and mounted the images side-by-side as a montage (*Figure 6C*). In that montage. we color-coded membranes according to their appearance (first generation: yellow, second: cyan, third: magenta, fourth: green). This

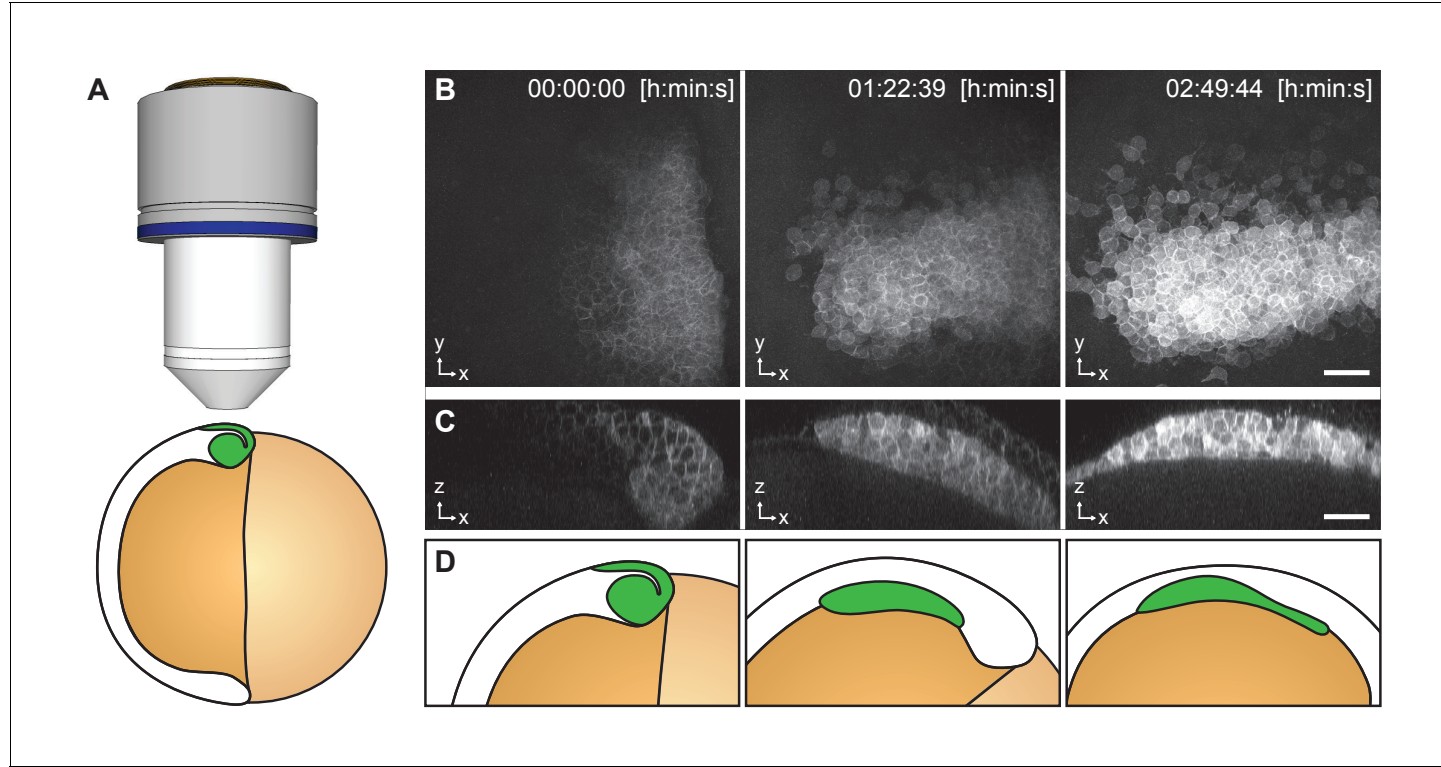

**Figure 9.** Imaging of Zebrafish prechordal plate using TipTracker with the LavisionBiotech TriM Scope II. (**A**) Schematic representation of the shield stage of a six hpf gsc::mEGFP Zebrafish embryo. Membrane bound EGFP is expressed in the prechordal plate (ppl) cells that form the shield (green). (**B**) Maximum intensity projections of three time points out of 78 show the gsc::mEGFP expression over time. (**C**) A transversal section (3.40 μm width maximum intensity projection) shows ppl cells ingression and subsequent migration. See also *Video 8*. (**D**) Schematic representation of gsc::mEGFP expressing cells ingression and migration between shield stage and 90% epiboly stage (9 h post fertilization). A stack of 50 images (x/y/z: 1024 × 1024 × 50 pixels, voxelsize: 0.342 × 0.342 × 3.0 μm$^3$) was captured every 2 min 15 s for a period of 2 h 50 min using the Zeiss Plan-Apochromat 20x/1.0 air objective lens. Time 0 corresponds to six hpf. Scale bars: (**B, C**) 50 μm.

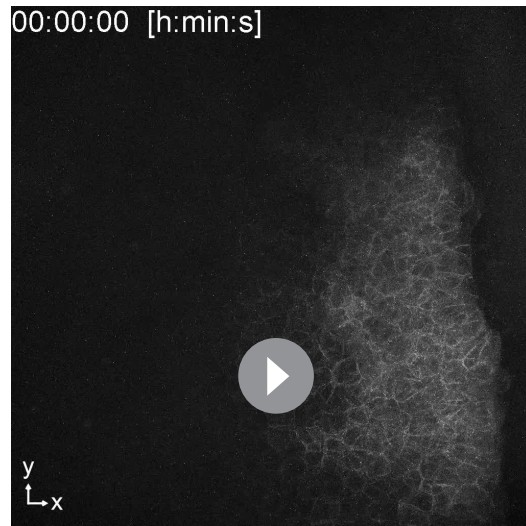

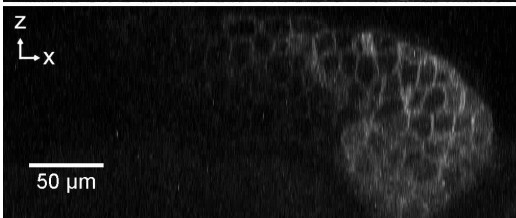

**Video 7.** Time series of Zebrafish prechordal plate using TipTracker with the LavisionBiotech TriM Scope.

experiment revealed unexpected regularity in the cell division pattern: Each first-generation membrane observed in the first time point is separated by three new membranes in the last time point of recording (*Figure 6C*, *Video 3*).

## Imaging of the KNOLLE syntaxin during cell division

To test the tracking using higher magnification, we analyzed the dynamics of the expression of the cell plate-specific syntaxin KNOLLE during cell division (*Figure 7* upper panel, *Video 4*) (*Lauber et al., 1997*; *Reichardt et al., 2007*). For this purpose, we used the Plan-Apochromat 40x/ 0.95 air objective lens, as the immersion liquid is not suitable for objectives in a horizontal position and multi-position acquisition. Still, with the air objective, we were able to follow the entire life cycle of the KNOLLE syntaxin that first localizes to the growing cell plate and after completion of cytokinesis relocalizes to pre-vacuolar compartments and is finally degraded in the vacuole, as described previously (*Reichardt et al., 2007*). In our setup, we could follow this cycle in a given cell and measured that it lasted for more than 4 h (*Figure 7*, lower panel). We successfully imaged growing roots for more than 12 h and captured a stack of 10 z-sections every 3 min using a 40x air objective. We further quantified the average duration of GFP-KNOLLE visibility in 83 cytokinesis events from a 24 h recording of six roots (*Video 5*, recording interval 15 min) and measured an average time period of 3 h 49 min ± 18 min. In five out of six roots we captured stem cell initials divisions. Remarkably, these cell divisions occurred within a time window of about 10 h, while before or after we did not observe any dividing initial stem cells. This suggests that stem cells synchronize their mitosis. It was reported that there is a modest correlation in timing of cell division between individual initials (*Campilho et al., 2006*). In our setup, compared to *Campilho et al. (2006)* who used a similar solution for tracking root tips following only a middle section of the root and for shorter periods of time, we captured the entire stem cell niche over prolonged periods of time. Our result supports and strengthens the hypothesis that stem cell initials divide together in a wave. We also can confirm that there is no correlation between day time and cell divisions, like it was reported for primary root tip (*Campilho et al., 2006*) or during lateral root formation (*von Wangenheim et al., 2016*).

## Imaging of the DII-Venus after gravistimulation

As a next example, to test how we can visualize dynamic processes during gravitropism, we observed roots during gravistimulation using the rotation stage. For this purpose, we used the DII-Venus auxin response marker line (*Brunoud et al., 2012*). A time series of vertically grown roots was recorded for 1 h, then gravistimulated by a 90° rotation, and finally turned back to the original position after 1.5 h (*Figure 8A*, *Video 6*). In total, we imaged three roots with a time interval of 3 min for a period of 4 h. During the rotation, imaging was stopped for the approximately 3 min that were necessary for the handling and starting of the new experiment. We then quantified the root tip angle and the DII-Venus fluorescence intensity gradient in the upper and lower parts of the roots before, during, and after gravistimulation (*Figure 8B*). We observed a rapid change in the angle that paralleled an increase in the fluorescence in the upper part of the root, and a fluorescence intensity decrease in the lower part. The upper/lower DII-Venus intensity ratio increased and reached a

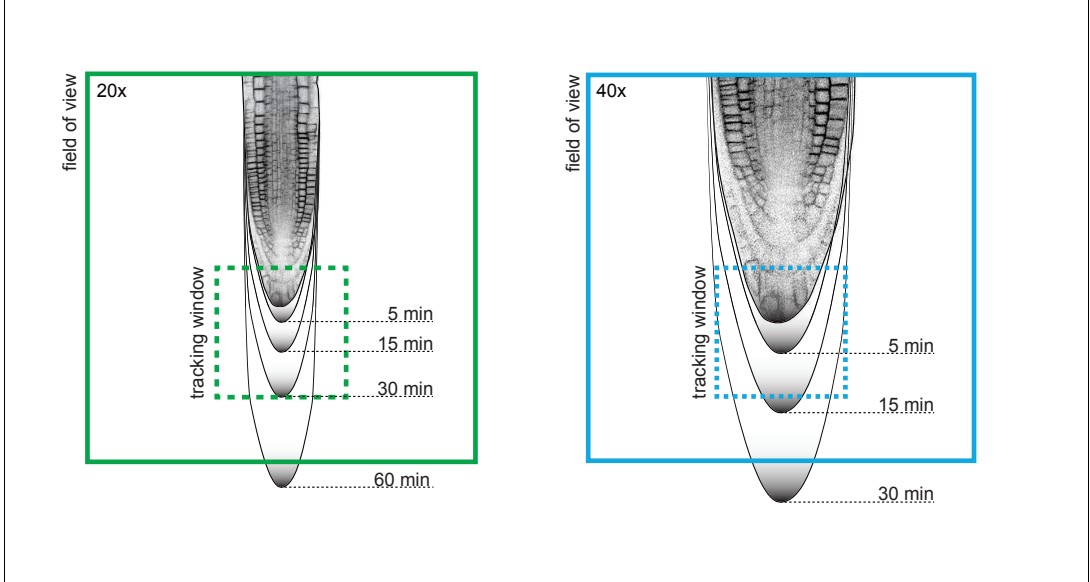

**Figure 10.** Root tip growth rate and tracking window. For successful tracking, it is recommended to use a spatio-temporal resolution in relation to the expected growth rate of the root. The maximum growth rate of *Arabidopsis* primary root tip can be 300 μm/h. Black gradient lines indicate how much a root tip might grow after certain periods of time. The dashed square indicates the tracking area which is 1/3 of field-of-view's dimensions (20x = 213 μm², 40x = 106 μm²).

plateau approximately 80 min after gravistimulation. This dynamics is similar to what has been described by *Band et al. (2012)*, but our setup provides higher temporal resolution compared to the periscope solution used before. After gravistimulation, we also observed the appearance of the DII-Venus signal in the columella, hinting to strong activation of auxin efflux from these cells. The second gravistimulation showed a rapid flux of auxin to the previously upper root side, causing a wave-like disappearance of the DII-Venus signal in these cells (*Video 6*). As seen in the movies, we were able to observe gradual changes in the DII-monitored auxin distribution from cell-to-cell with previously unseen spatial and temporal resolution.

## Imaging of the moving prechordal plate in developing zebrafish embryo

To demonstrate that TipTracker can be used on completely different microscope setups, that are moreover in the standard upright/inverted configuration, and for non-plant samples, we visualized and tracked the movement of the prechordal plate in a developing zebrafish embryo with the LavisionBiotech TriM Scope II. The anterior axial mesendoderm cells segregate from the ectoderm progenitor cells via synchronized cell ingression (*Montero et al., 2005*). Once ingressed, they form a compact cell cluster, the prospective prechordal plate (ppl), and collectively migrate toward the animal pole of the gastrula (*Montero et al., 2003*, *2005*; *Dumortier et al., 2012*). Zebrafish embryos have an average diameter of 600 μm and the ppl migrates along the circumference of the embryo and reaches the animal pole in about 4 h. Ppl cell fate specification can be visualized by the expression of the *goosecoid* (*gsc*) marker gene (*Schulte-Merker et al., 1994*). We used the zebrafish transgenic line expressing *gsc::mEGFP* (*Smutny et al., 2017*), to analyze gsc expression as well as cell shape and movement of ppl cells in vivo. As shown in *Figure 9* and *Video 7*, using TipTracker we were able to follow the cluster of ppl cells, enabling uninterrupted imaging for several hours.

These examples of the performance of TipTracker show that the program can be used to track root tips at high temporal and spatial resolution; we used both the 20x and 40x objectives and timeframes ranging from 3 to 20 min. In addition to root tips, it is also possible to track other moving samples on non-vertical microscope setups, as we demonstrate with the example of the prechordal plate movement in the zebrafish embryo. It is important to note that when setting up an experiment, the users should consider the magnitude of the velocity of their sample relative to the field of view of the objective being used, as well as to the temporal resolution of the acquisition (*Figure 10*). For

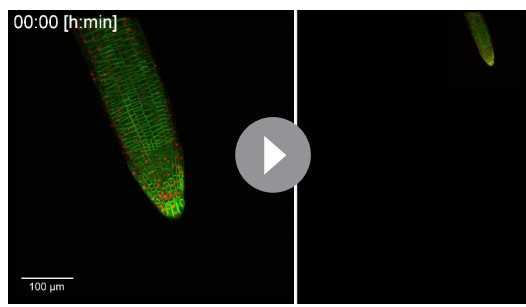

**Video 8.** Time series of a root tip growing along another root demonstrating the robustness of tracking in a spectacular way.

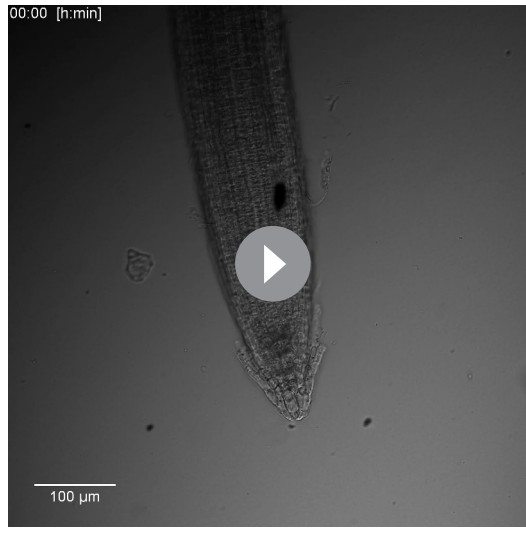

**Video 9.** Time series of a root tip growing away from the objective lens inside the gel.     As a result, the root tip is no more in focus but the blurred root is still tracked by TipTracker.

example, when using a 20x objective lens, we recommend specifying time intervals not larger than 30 min, otherwise the root will escape the tracking field of view before the next time point is captured.

## Conclusions and discussion

In this work, we describe in detail a confocal microscope setup with a vertical mounting. This enables long-term (up to several days) live imaging with confocal resolution of seedlings growing in the natural, vertical position. We also built a rotation stage that makes it possible to freely adjust the plant's orientation with respect to gravity, while preserving the ability to observe it. Together with integrated illumination, our setup provides growing seedlings with the optimal and controlled conditions necessary for long-term imaging experiments. We provide blueprints for building the setup and a description of optimized sample preparation, which is a critical step for the sensitive *Arabidopsis* seedlings. Together, the sample preparation, illumination and the vertical position result in healthy seedlings, even in the artificial conditions of a confocal microscope.

Furthermore, we developed the TipTracker program to automatically follow root tips for long periods of time. Importantly, it can track multiple objects simultaneously while fully preserving the functionality of the confocal, that is multiple-color imaging and z-sections. Brightfield and fluorescence channels can be used as the input for tracking. The tracking is both robust and very accurate, as exemplified by *Videos 8* and *9*. TipTracker tracks objects only in two dimensions since the roots are confined between the coverslip and the agar block, but if needed, 3D tracking is a straightforward extension.

In addition to confocal microscopes, the lightsheet (LSFM, SPIM) systems are ideally suited for the imaging of roots with TipTracker because these usually already feature vertical sample mounting. As an example, the implementation of TipTracker on a Zeiss Z1 lighsheet system can be achieved rather straightforward, however, image file handling and position input would need to be modified. For home-built SPIM systems with custom software, it is easier to directly implement the tracking calculation in the acquisition software, here the TipTracker's core algorithms can be used.

The usage of TipTracker is not limited to vertical stage microscopes and can be used on any inverted or upright microscope setup with a motorized stage. The example of zebrafish embryo development also demonstrates that the tracking algorithm is not limited to root tips and can in fact be used for all moving samples, given that the algorithm makes no assumptions about the samples, such as shape, brightness, or direction of movement.

Combining optimal growing conditions and root tip tracking, we now have been able to perform experiments that were previously very hard to conduct. Long-term image acquisition revealed previously unappreciated regularity in the cell division and elongation pattern making up the root growth. The high-resolution imaging of dividing cells enabled capturing the exact timing of several

cytokinesis events while observing the whole root meristem; or the observation of the dynamic rearrangement of the auxin gradient during gravitropism making it possible to dissect the spread of auxin distribution at high spatio-temporal resolution. Our setup made these findings possible, demonstrating its versatility and application to a broad range of questions in developmental, cell biology and physiology. We have aimed for a very detailed description that will enable other labs to implement the setup or its components, and will therefore be beneficial to the *Arabidopsis* community as well as non-plant researchers.

## Material and methods

### Plant transgenic reporter lines
The transgenic reporter lines were described previously: KNOLLE::GFP-KNOLLE (*Reichardt et al., 2007*), DII-Venus (*Brunoud et al., 2012*), UBQ10::H2B-RFP / UBQ10::YFP-PIP1;4 / GATA23::nls-GUS-GFP (*von Wangenheim et al., 2016*).

### Illumination setup
The LED illumination system is a custom-built lamp. The design of the illumination lamp was drawn using a PCB-software (PCB: printed circuit board). We provide the board design file in the supplemental material. The board was then manufactured and assembled in our institute's machine shop. Each of the four boards is able to accommodate 16 LEDs, we equipped it with five pairs of red and blue LEDs (blue LED: 453 nm, OSRAM LD CN5M-1R1S-35–1, and red LED: 625 nm, OSRAM LR T66F-ABBB-1–1). The voltage can be adjusted in the range of 3.5–9.5 V. Appropriate resistors were used to reach light intensities ranging from 40 to 180 $\mu$mol/m$^2$/s (see *Figure 3*).

### Plant sample preparation
Plants were surface sterilized by chlorine gas, sowed on ½ MS medium, 1% sucrose, pH 5.8, 0.8% plant agar, stratified 1–2 days at 4°C and then cultivated in a growth incubator at 22°C in a 16/8 h day/night cycle with 120–140 $\mu$mol/m$^2$/s amount of light for 5 days.

Sample preparation protocol for Lab-Tek chambered coverglass (Thermo Scientific Nunc, catalogue no: 15536)

1. Prepare ½ MS medium (half-strength Murashige and Skoog medium) by adding 2.15 g MS-medium, 10 g sucrose, 0.97 g MES (2-(N-morpholino)ethanesulfonic acid) and 1 L ddH2O (double-distilled water) into a 1 L bottle. Adjust the pH to 5.8 using KOH.
2. Add 15 g/L phytagel or agar to the ½ MS medium and autoclave it for 20 min at 121°C.
3. Pour 50 ml of the hot medium into square petri dishes (245 $\times$ 245 $\times$ 25 mm) and let it cool down to room temperature to allow the medium to solidify.
4. Cut a block of gel that fits into the chambered coverslip. Therefore, stamp the chamber upside down into the gel. Note: Do not push it into the gel, just mark the boundaries of the chamber on the gel surface. Then cut the gel using a scalpel and remove a 2 mm stripe along the long side, this is where the leaves will find space.
5. Transfer *Arabidopsis* seedlings carefully on the block of gel. Work uninterruptedly and try to avoid any air draft (rapid movements, air-condition flow).
6. Lift the block of gel using a spatula and slide it into the chamber such that the plants are between gel and glass. Note: Try to avoid air bubbles.
7. Close the lid and wrap the chamber with thin tape.
8. Cultivate the chamber in a growth incubator, e.g. at 22 °C in a 16/8 h day/night cycle with 120–140 $\mu$mol/m$^2$/s amount of light. In order to let the plants acclimate and recover from stress of transfer cultivate them for at least 1 h before imaging.

### Zebrafish embryo imaging
Fish maintenance and embryo collection were carried out as previously described (*Westerfield, 2007*). Embryos were raised in either E3 medium or Danieau's buffer, kept at 28 or 31°C and staged according to *Kimmel et al. (1995)*. To analyze the movement of ppl cells expressing gsc::mEGFP, embryos were kept at 31°C until shield stage (6 h post fertilization). Embryos were dechorionated with forceps, mounted in 0.7% agarose in E3 medium and imaged with a LaVision upright

multi-photon microscope equipped with a Zeiss Plan-Apochromat 20x/1.0 water immersion objective and Ti:Sa laser (Chameleon, Coherent) set at 820 nm. The xy position of the motorized stage was adjusted automatically after every acquisition of a 50 images stack (z step 3 μm).

### Image analysis

Cell division analysis in *Figure 6*: A single z-section of data set number #05 was stabilized around one cell file using semi-automatic motion tracking in Adobe After Effects. The image sequence was exported as tif files and imported into Fiji. The area, highlighted with a dashed white box in *Figure 6C* (right), was cut out and a montage of each time point was created using the Fiji function 'Make Montage'. In AdobeIllustrator membranes were labeled using the path tool, the transparency mode was set to color. For the *Video 3*, the colored image was imported into Fiji and the montage was reversed into a stack of individual time points using the function 'Montage to Stack'.

Angle measurement and fluorescent intensity measurement in *Figure 8*: Sum intensity projections of each time point were done in Fiji. Then the time series was stabilized using the 'Linear Stack Alignment with SIFT' function in Fiji. A rectangle area (450 × 691 pixels) was drawn overlapping with one side of the root tip and the fluorescence intensity was measured at each time point. Angles were calculated from coordinates of two points, the tip of the root and the position of the organizing center, obtained from semi-automatic motion tracking in Adobe After Effects.

## Acknowledgements

The authors are grateful to the Miba Machine Shop at IST Austria for their contribution to the microscope setup and to Yvonne Kemper for reading, understanding and correcting the manuscript. This work was supported by the Austrian Science Fund (FWF01_I1774S) to EB.

## Additional information

### Funding

| Funder | Grant reference number | Author |
| --- | --- | --- |
| Marie Curie Actions | FP7/2007-2013 no 291734 | Daniel von Wangenheim |
| Austrian Science Fund | M 2128-B21 | Matyáš Fendrych |
| Austrian Science Fund | FWF01_I1774S | Eva Benková |
| European Research Council | FP7/2007-2013 no 282300 | Jiří Friml |

The funders had no role in study design, data collection and interpretation, or the decision to submit the work for publication.

### Author contributions

DvW, Conceptualization, Data curation, Validation, Investigation, Visualization, Methodology, Writing—original draft, Writing—review and editing; RH, Conceptualization, Software, Validation, Investigation, Methodology, Writing—original draft, Writing—review and editing; MF, Conceptualization, Data curation, Funding acquisition, Validation, Investigation, Methodology, Writing—original draft, Writing—review and editing; VB, Validation, Investigation, Methodology, Writing—review and editing; EB, Conceptualization, Funding acquisition, Writing—review and editing; JF, Conceptualization, Funding acquisition, Writing—original draft, Writing—review and editing

### Author ORCIDs

Daniel von Wangenheim, http://orcid.org/0000-0002-6862-1247
Robert Hauschild, http://orcid.org/0000-0001-9843-3522
Matyáš Fendrych, http://orcid.org/0000-0002-9767-8699
Jiří Friml, http://orcid.org/0000-0002-8302-7596

# Additional files

**Supplementary files**

• Supplementary file 1. Hardware. (1) Board design of the illumination system. (2) Mounting plate for Axio Observer. (3) Modification of the laser safety. (4) Rotation stage inset.

• Supplementary file 2. Collection of all scripts and supporting information. (1) Implementation of TipTracker on two commercial platforms (Zeiss LSM700 and LaVisionBiotec TriMScopeII) and a short manual how to use it. (2) Fiji macros to convert LSM files into Hyperstacks. (3) Collection of simple AutoIt scripts and description on how to adapt them to a specific setup. (4) Script to calculate a post-rotation position list to use with the rotation stage.

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
