## [Decision Letter]

Thank you for submitting your article "Live tracking of moving samples in confocal microscopy for vertically grown plant roots" for consideration by *eLife*. Your article has been reviewed by three peer reviewers, and the evaluation has been overseen by a Reviewing Editor and Christian Hardtke as the Senior Editor.

The reviewers have discussed the reviews with one another and the Reviewing Editor has drafted this decision to help you prepare a revised submission.

In principal, the reviewers found the set-up reported in your manuscript useful. However, they are concerned about its scope as a resource at this point and would like to see further improvements and documentation:

While there was consensus that a resource should be rather easily implemented/applied, the conversion to a vertical microscope is not straightforward (and probably would result in loss of warranty for confocal systems). Yet, as this is of course unavoidable here, at least the tracking part should become easier to implement and we therefore would like to ask you to provide the software in an open format.

Please discuss the limitations of the set up in more depth and make each aspect very transparent such that it is truly informative for the community.

An important point is that the resource should demonstrate that its use leads to new insight into root development beyond what was previously possible. This might be achieved by a more in depth analysis of KNOLLE behavior, or by additional pertinent experiments. For example, can it be used to follow markers in inner tissue layers across time?

For details, refer to the reviews below, and please also address all other points brought up by the reviewers.

Reviewer #1:

This is an exciting and innovative manuscript which addresses a key imaging obstacle in plant biology. While other solutions to the extended imaging of gravity-sensitive roots have been provided previously, this work provides significant advances beyond these enabling this to be achieved for far greater time periods.

The text is clearly written and the methods are well described. The quality of the videos provided is outstanding.

Concerns with the manuscript lie principally within the experimental validation of the technology, and a couple with the method itself.

In order to encourage this technology to be adopted, it would be preferred to have the software work in the freely available Octave software as opposed to MatLab. This would remove a financial obstacle in the form of licence-based software. In a best case scenario the existing code will run seamlessly within Octave.

The extent to which plants are experiencing a hypoxic response under agar within the Lab-Tek chambered coverglass should be examined. Imaging the ADH1::GFP reporter would answer this. Knowing whether plants are experiencing hypoxia will impact the interpretation of data obtained using this technology.

More detail on the novel insights arising from imaging KNOLLE would strengthen this part of the text.

DII and gravity stimulation has been performed previously using a periscope in the PNAS paper "Root gravitropism is regulated by a transient lateral auxin gradient controlled by a tipping-point mechanism". This and differences between what is observed here and what has been reported previously requires further detailed discussion.

Verification of cell tracking in zebrafish through some form of quantification is lacking. A clear validation of the method and the ability to derive novel insight into this biological process is missing.

Reviewer #2:

Von Wangenheim and coworkers describe a vertical confocal setup and developed a custom software for automatic tracking of moving/growing objects. The authors very nicely explain how to convert a "normal" confocal into a vertical one. They have supplied all the needed information for the transformation and for the construction of the rotation stage, illumination setup and the scripts to implement the TipTracker software. They have used this system to show that it can give high resolution information on cell differentiation, division and gravitropic responses. Although all experiments are performed to the highest standards and the data is represented in a beautiful and clear manner, to me it falls short to be considered as a resource. The conversion to a vertical confocal system is not straightforward and needs experts in optics and also experienced workshops. This is not readily available for most laboratories. Also when comparing the manuscript to other resource papers such as Barbier de Reuille et al., 2015 and Rabe et al., 2016 this manuscript will have less impact on the community as in my opinion is it limited to only a few groups. It might be an idea to expand the Tiptracker idea also to light sheet microscope setups. I feel that in its current state it falls short of being a resource for the community.

Reviewer #3:

The authors present a methodology how to convert an upright confocal microscope into vertical imaging system where the growth and movement of biological samples can be tracked and recorded in a high resolution.

Imaging in vertical orientation facilitates studying of gravity responding samples in their natural orientation e.g. plant root tips.

The article describes in detail how to adapt the upright microscope for its new function. Additionally, the authors provide a tracking program that enables the process of long time-course scanning to be automated for several samples at the same time. Importantly, the authors provide also the information on how to provide good light necessary for undisturbed growth.

This methodology will facilitate more accurate studies of gravity influenced processes in the growing root.

---

## [Author Response]

*While there was consensus that a resource should be rather easily implemented/applied, the conversion to a vertical microscope is not straightforward (and probably would result in loss of warranty for confocal systems). Yet, as this is of course unavoidable here, at least the tracking part should become easier to implement and we therefore would like to ask you to provide the software in an open format.*

Even in the vertical configuration the confocal scan head (the actual core of the ‘confocal’) is operating in the same orientation as in the inverted setup as the microscope body and not the scan head is turned by 90°. The confocal scan head is not modified in any way. Therefore, we see no reason why this would affect the warranty. The situation concerning the microscope body is indeed different: No commercial manufacturer designs and tests its microscopes in the vertical configuration. In this version of the manuscript (in the TipTracker part) we list the prerequisites of the setup to be verticalized.

We added a compiled version of the TipTracker program, which enables the users who do not need to modify the code to run it without a license. But we argue that MATLAB has several crucial advantages over the freeware software solutions; mainly the graphical user interface that in fact crucially facilitates the implementation of TipTracker. Please see our detailed response to the ‘open format’ issue with MATLAB under the respective reviewer’s comment

*Please discuss the limitations of the set up in more depth and make each aspect very transparent such that it is truly informative for the community.*

We tried to make the limitations of the system more clear in this version of the manuscript.

*An important point is that the resource should demonstrate that its use leads to new insight into root development beyond what was previously possible. This might be achieved by a more in depth analysis of KNOLLE behavior, or by additional pertinent experiments. For example, can it be used to follow markers in inner tissue layers across time?*

We performed an additional experiment with the KNOLLE marker line, see the details below under the respective comment. It is important to note that our setup does not in any way change the original properties of the given microscope setup; therefore imaging of inner tissues will be similar on the vertical or horizontal microscopes. The major advantage is the ability to image over time without the need of the user to reposition the sample; therefore we focused on the temporal dimension of the KNOLLE behavior in this version of the manuscript.

*Reviewer #1:*

*Concerns with the manuscript lie principally within the experimental validation of the technology, and a couple with the method itself.*

With the presubmission to Biorxiv all software has already been provided as open source licensed under CC BY-NC 4.0 (creative commons, Attribution-NonCommercial 4.0). This means that all our software can freely be copied and redistributed in any medium or format, as well as remixed, transformed and built upon. Part of the software is written in Matlab which is indeed a proprietary programming language. However, we cannot agree that this would hamper the implementation of TipTracker:

1) Since the entire TipTracker software is already open source, anyone is free to port it to her preferred programming language such as Python, Java (for Micromanager), Octave etc. With regards to Octave as an alternative to Matlab, Matlab was chosen because it has several advantages: Matlab offers solid, well tested and documented functionality as well as a great debugging environment. Octave is principally a batch or command-line language. While it is theoretically possible to create a graphical user interface (GUI) with octave, doing so would add a lot of complexity to the code. The code we provide is intentionally kept as basic as possible in order to facilitate modification and extension. In contrast to Octave, Matlab has “guide” that makes building GUI front ends very easy. We felt that having a point-and-click control for TipTracker and thus eliminating the need to type commands in order to run the application was important for the user acceptance.

2) No Matlab license is required to run TipTracker as is. We added a compiled version of TipTracker, which enables royalty-free deployment to users who do not need to modify the code.

3) Investment as well as running costs are about two orders of magnitude higher for a confocal microscope than for Matlab, hence the requirement of having a motorized microscope and not Matlab is the real bottleneck for adaptation.

*In order to encourage this technology to be adopted, it would be preferred to have the software work in the freely available Octave software as opposed to MatLab. This would remove a financial obstacle in the form of licence-based software. In a best case scenario the existing code will run seamlessly within Octave.*

*The extent to which plants are experiencing a hypoxic response under agar within the Lab-Tek chambered coverglass should be examined. Imaging the ADH1::GFP reporter would answer this. Knowing whether plants are experiencing hypoxia will impact the interpretation of data obtained using this technology.*

We ordered the ADH::GFP marker line and tested it in the setup and we never observed induction of GFP signal. However, we failed to induce its expression also by other means (24h submersion in water, mild vacuum, removal of oxygen by oxygen absorbers, which actually lead to growth arrest), and we lack the sophisticated tools to precisely control oxygen content in the air. Therefore we do not feel comfortable showing such negative data without a proper positive control. Instead, we compared the growth of roots in the chamber between the block of agar and the coverslip with the growth of roots in the same chamber on the surface of the agar block. Roots between agar and glass grew 88% of the ones growing on the surface (87.94% +-10.5% STDEV, control 100% +-10.9% STDEV, n=21 and 20, respectively), which can be attributed to the mechanical impedance of the setup. We added this analysis into the text.

*More detail on the novel insights arising from imaging KNOLLE would strengthen this part of the text.*

We performed an additional 24-hour imaging of GFP-KNOLLE marker line and quantified the duration of GFP-KNOLLE appearance during cytokinesis (83 cytokinesis events) and we focused on the stem cell divisions. We added an additional supplemental video and describe the results in the text.

*DII and gravity stimulation has been performed previously using a periscope in the PNAS paper "Root gravitropism is regulated by a transient lateral auxin gradient controlled by a tipping-point mechanism". This and differences between what is observed here and what has been reported previously requires further detailed discussion.*

We included the discussion of this paper into our text, reanalyzed the data to be comparable to the Band et al., 2012 paper and we modified the figure accordingly.

*Verification of cell tracking in zebrafish through some form of quantification is lacking. A clear validation of the method and the ability to derive novel insight into this biological process is missing.*

We have to respectfully disagree here: This experiment was done to demonstrate that TipTracker is able to run on a completely different setup than a vertical confocal and is able to track the movement of a sample that is not a root tip. This shows that users can use TipTracker without the need of verticalizing their microscopes, with minimal interference with the existing setups. We believe that tracking animal cells on a horizontal microscope is an ideal demonstration of this ability.

*Reviewer #2:*

*Von Wangenheim and coworkers describe a vertical confocal setup and developed a custom software for automatic tracking of moving/growing objects. The authors very nicely explain how to convert a "normal" confocal into a vertical one. They have supplied all the needed information for the transformation and for the construction of the rotation stage, illumination setup and the scripts to implement the TipTracker software. They have used this system to show that it can give high resolution information on cell differentiation, division and gravitropic responses. Although all experiments are performed to the highest standards and the data is represented in a beautiful and clear manner, to me it falls short to be considered as a resource. The conversion to a vertical confocal system is not straightforward and needs experts in optics and also experienced workshops. This is not readily available for most laboratories. Also when comparing the manuscript to other resource papers such as Barbier de Reuille et al., 2015 and Rabe et al., 2016 this manuscript will have less impact on the community as in my opinion is it limited to only a few groups. It might be an idea to expand the Tiptracker idea also to light sheet microscope setups. I feel that in its current state it falls short of being a resource for the community.*

We thank the reviewer for appreciating the quality of our experiments. We however cannot agree that our manuscript should not be considered a resource. We believe that our manuscript is in fact a collection of tools, and as such fits the “tools and resources” category of *eLife* articles. We share tools and protocols with the community that can be used altogether as we did, but also each of them separately:

Detailed description of the ‘verticalization’ of a confocal microscope

Detail description and blueprints for plant illumination for a microscope

Detailed description of a microscope rotation inset

The TipTracker software that enables live tracking of roots, but also other moving samples.

Detailed protocol of Arabidopsis seedling sample preparation for long-term live imaging

Of course the confocal microscope is an expensive tool and therefore needs to be handled by experts, but the conversion of an inverted microscope to a vertical system involves no modification of the optics. Likewise, neither the microscope body, nor the confocal scanhead are altered in any way. Solely, the mounting is changed: The microscope body is screwed onto a custom plate and the scan head and the transmitted light arm are raised by support platforms. As platforms simple commercial labjacks suffice. Since we provide a CAD file for the mounting plate it can be readily ordered from any third party machine shop. The modification of the laser safety is a matter of convenience and furthermore quite simple.

With the example of zebrafish embryogenesis, we were aiming to demonstrate that TipTracker can be implemented on ANY microscope with a motorized stage with controllable positions, not only to verticalized setups. Also any live root imaging longer than 20-30 minutes suffers from the growth of the root tip, given that the sample is in a good condition. Therefore in case of imaging roots in a horizontal position the tracking will be more than useful, and can be implemented without any modifications to existing setups.

The usage on lightsheets – SPIMs is a good point: SPIM systems usually already feature vertical sample mounting. Therefore they are ideally suited for the imaging of roots with TipTracker. As an example, adapting TipTracker to a Zeiss Z1 lighsheet system can be achieved however image file handling and position input would require modification. For home-build SPIM systems with custom software it is easier to directly implement the tracking calculation in the acquisition software, but for this the tracking algorithm that is in the TipTracker’s core can be used.

We discuss the above-mentioned points at the end of the TipTracker chapter of the manuscript.